# Discrete Graph Hashing

**Wei Liu**[†]    **Cun Mu**[‡]    **Sanjiv Kumar**[♯]    **Shih-Fu Chang**[‡]
[†]IBM T. J. Watson Research Center    [‡]Columbia University    [♯]Google Research
weiliu@us.ibm.com    cm3052@columbia.edu
sfchang@ee.columbia.edu    sanjivk@google.com

## Abstract

Hashing has emerged as a popular technique for fast nearest neighbor search in gigantic databases. In particular, learning based hashing has received considerable attention due to its appealing storage and search efficiency. However, the performance of most unsupervised learning based hashing methods deteriorates rapidly as the hash code length increases. We argue that the degraded performance is due to inferior optimization procedures used to achieve discrete binary codes. This paper presents a graph-based unsupervised hashing model to preserve the neighborhood structure of massive data in a discrete code space. We cast the graph hashing problem into a discrete optimization framework which directly learns the binary codes. A tractable alternating maximization algorithm is then proposed to explicitly deal with the discrete constraints, yielding high-quality codes to well capture the local neighborhoods. Extensive experiments performed on four large datasets with up to one million samples show that our discrete optimization based graph hashing method obtains superior search accuracy over state-of-the-art unsupervised hashing methods, especially for longer codes.

## 1   Introduction

During the past few years, hashing has become a popular tool for tackling a variety of large-scale computer vision and machine learning problems including object detection [6], object recognition [35], image retrieval [22], linear classifier training [19], active learning [24], kernel matrix approximation [34], multi-task learning [36], *etc*. In these problems, hashing is exploited to map similar data points to adjacent binary hash codes, thereby accelerating similarity search via highly efficient Hamming distances in the code space. In practice, hashing with short codes, say about one hundred bits per sample, can lead to significant gains in both storage and computation. This scenario is called *Compact Hashing* in the literature, which is the focus of this paper.

Early endeavors in hashing concentrated on using random permutations or projections to construct randomized hash functions. The well-known representatives include Min-wise Hashing (MinHash) [3] and Locality-Sensitive Hashing (LSH) [2]. MinHash estimates the Jaccard set similarity and is improved by $b$-bit MinHash [18]. LSH can accommodate a variety of distance or similarity metrics such as $\ell_p$ distances for $p \in (0, 2]$, cosine similarity [4], and kernel similarity [17]. Due to randomized hashing, one needs more bits per hash table to achieve high precision. This typically reduces recall, and multiple hash tables are thus required to achieve satisfactory accuracy of retrieved nearest neighbors. The overall number of hash bits used in an application can easily run into thousands.

Beyond the data-independent randomized hashing schemes, a recent trend in machine learning is to develop data-dependent hashing techniques that learn a set of compact hash codes using a training set. Binary codes have been popular in this scenario for their simplicity and efficiency in computation. The compact hashing scheme can accomplish almost constant-time nearest neighbor search, after encoding the whole dataset to short binary codes and then aggregating them into a hash table. Additionally, compact hashing is particularly beneficial to storing massive-scale data. For example, saving one hundred million samples each with 100 binary bits costs less than 1.5 GB, which

can easily fit in memory. To create effective compact codes, several methods have been proposed. These include the unsupervised methods, *e.g.*, Iterative Quantization [9], Isotropic Hashing [14], Spectral Hashing [38, 37], and Anchor Graph Hashing [23], the semi-supervised methods, *e.g.*, Weakly-Supervised Hashing [25], and the supervised methods, *e.g.*, Semantic Hashing [30], Binary Reconstruction Embeddings [16], Minimal Loss Hashing [27], Kernel-based Supervised Hashing [22], Hamming Distance Metric Learning [28], and Column Generation Hashing [20].

This paper focuses on the problem of unsupervised learning of compact hash codes. Here we argue that most unsupervised hashing methods suffer from inadequate search performance, particularly low recall, when applied to learn relatively longer codes (say around 100 bits) in order to achieve higher precision. The main reason is that the discrete (binary) constraints which should be imposed on the codes during learning itself have not been treated adequately. Most existing methods either neglect the discrete constraints like PCA Hashing and Isotropic Hashing, or discard the constraints to solve the relaxed optimizations and afterwards round the continuous solutions to obtain the binary codes like Spectral Hashing and Anchor Graph Hashing. Crucially, we find that the hashing performance of the codes obtained by such relaxation + rounding schemes deteriorates rapidly when the code length increases (see Fig. 2). Till now, very few approaches work directly in the discrete code space. Parameter-Sensitive Hashing [31] and Binary Reconstruction Embeddings (BRE) learn the parameters of predefined hash functions by progressively tuning the codes generated by such functions; Iterative Quantization (ITQ) iteratively learns the codes by explicitly imposing the binary constraints. While ITQ and BRE work in the discrete space to generate the hash codes, they do not capture the local neighborhoods of raw data in the code space well. ITQ targets at minimizing the quantization error between the codes and the PCA-reduced data. BRE trains the Hamming distances to mimic the $\ell_2$ distances among a limited number of sampled data points, but could not incorporate the entire dataset into training due to its expensive optimization procedure.

In this paper, we leverage the concept of Anchor Graphs [21] to capture the neighborhood structure inherent in a given massive dataset, and then formulate a graph-based hashing model over the whole dataset. This model hinges on a novel discrete optimization procedure to achieve nearly balanced and uncorrelated hash bits, where the binary constraints are explicitly imposed and handled. To tackle the discrete optimization in a computationally tractable manner, we propose an alternating maximization algorithm which consists of solving two interesting subproblems. For brevity, we call the proposed discrete optimization based graph hashing method as *Discrete Graph Hashing* (DGH). Through extensive experiments carried out on four benchmark datasets with size up to one million, we show that DGH consistently obtains higher search accuracy than state-of-the-art unsupervised hashing methods, especially when relatively longer codes are learned.

## 2 Discrete Graph Hashing

First we define a few main notations used throughout this paper: $\mathrm{sgn}(x)$ denotes the sign function which returns 1 for $x > 0$ and $-1$ otherwise; $\mathbf{I}_n$ denotes the $n \times n$ identity matrix; $\mathbf{1}$ denotes a vector with all 1 elements; $\mathbf{0}$ denotes a vector or matrix of all 0 elements; $\mathrm{diag}(\boldsymbol{c})$ represents a diagonal matrix with elements of vector $\boldsymbol{c}$ being its diagonal entries; $\mathrm{tr}(\cdot)$, $\| \cdot \|_{\mathrm{F}}$, $\| \cdot \|_1$, and $\langle \cdot, \cdot \rangle$ express matrix trace norm, matrix Frobenius norm, $\ell_1$ norm, and inner-product operator, respectively.

**Anchor Graphs.** In the discrete graph hashing model, we need to choose a neighborhood graph that can easily scale to massive data points. For simplicity and efficiency, we choose Anchor Graphs [21], which involve no special indexing scheme but still have linear construction time in the number of data points. An anchor graph uses a small set of $m$ points (called *anchors*), $\mathcal{U} = \{ \boldsymbol{u}_j \in \mathbb{R}^d \}_{j=1}^m$, to approximate the neighborhood structure underlying the input dataset $\mathcal{X} = \{ \boldsymbol{x}_i \in \mathbb{R}^d \}_{i=1}^n$. Affinities (or similarities) of all $n$ data points are computed with respect to these $m$ anchors in linear time $O(dmn)$ where $m \ll n$. The true affinity matrix $\mathbf{A}^o \in \mathbb{R}^{n \times n}$ is then approximated by using these affinities.

Specifically, an anchor graph leverages a nonlinear data-to-anchor mapping ($\mathbb{R}^d \mapsto \mathbb{R}^m$) $\boldsymbol{z}(\boldsymbol{x}) = \left[ \delta_1 \exp(-\frac{\mathcal{D}^2(\boldsymbol{x}, \boldsymbol{u}_1)}{t}), \cdots, \delta_m \exp(-\frac{\mathcal{D}^2(\boldsymbol{x}, \boldsymbol{u}_m)}{t}) \right]^\top / M$, where $\delta_j \in \{1, 0\}$ and $\delta_j = 1$ if and only if anchor $\boldsymbol{u}_j$ is one of $s \ll m$ closest anchors of $\boldsymbol{x}$ in $\mathcal{U}$ according to some distance function $\mathcal{D}()$ (*e.g.*, $\ell_2$ distance), $t > 0$ is the bandwidth parameter, and $M = \sum_{j=1}^m \delta_j \exp(-\frac{\mathcal{D}^2(\boldsymbol{x}, \boldsymbol{u}_j)}{t})$ leading to $\| \boldsymbol{z}(\boldsymbol{x}) \|_1 = 1$. Then, the anchor graph builds a data-to-anchor affinity matrix $\mathbf{Z} =$

$\left[\boldsymbol{z}(\boldsymbol{x}_1), \cdots, \boldsymbol{z}(\boldsymbol{x}_n)\right]^{\top} \in \mathbb{R}^{n \times m}$ that is highly sparse. Finally, the anchor graph gives a data-to-data affinity matrix as $\mathbf{A} = \mathbf{Z}\boldsymbol{\Lambda}^{-1}\mathbf{Z}^{\top} \in \mathbb{R}^{n \times n}$ where $\boldsymbol{\Lambda} = \mathrm{diag}(\mathbf{Z}^{\top}\mathbf{1}) \in \mathbb{R}^{m \times m}$. Such an affinity matrix empirically approximates the true affinity matrix $\mathbf{A}^o$, and has two nice characteristics: 1) $\mathbf{A}$ is a low-rank positive semidefinite (PSD) matrix with rank at most $m$, so the anchor graph does not need to compute it explicitly but instead keeps its low-rank form and only saves $\mathbf{Z}$ and $\boldsymbol{\Lambda}$ in memory; 2) $\mathbf{A}$ has unit row and column sums, so the resulting graph Laplacian is $\mathbf{L} = \mathbf{I}_n - \mathbf{A}$. The two characteristics permit convenient and efficient matrix manipulations upon $\mathbf{A}$, as shown later on. We also define an anchor graph affinity function as $A(\boldsymbol{x}, \boldsymbol{x}') = \boldsymbol{z}^{\top}(\boldsymbol{x})\boldsymbol{\Lambda}^{-1}\boldsymbol{z}(\boldsymbol{x}')$ in which $(\boldsymbol{x}, \boldsymbol{x}')$ is any pair of points in $\mathbb{R}^d$.

**Learning Model.** The purpose of unsupervised hashing is to learn to map each data point $\boldsymbol{x}_i$ to an $r$-bit binary hash code $\boldsymbol{b}(\boldsymbol{x}_i) \in \{1, -1\}^r$ given a training dataset $\mathcal{X} = \{\boldsymbol{x}_i\}_{i=1}^n$. For simplicity, let us denote $\boldsymbol{b}(\boldsymbol{x}_i)$ as $\boldsymbol{b}_i$, and the corresponding code matrix as $\mathbf{B} = [\boldsymbol{b}_1, \cdots, \boldsymbol{b}_n]^{\top} \in \{1, -1\}^{n \times r}$. The standard graph-based hashing framework, proposed by [38], aims to learn the hash codes such that the neighbors in the input space have small Hamming distances in the code space. This is formulated as:

$$\min_{\mathbf{B}} \quad \frac{1}{2}\sum_{i,j=1}^{n} \|\boldsymbol{b}_i - \boldsymbol{b}_j\|^2 A_{ij}^o = \mathrm{tr}(\mathbf{B}^{\top}\mathbf{L}^o\mathbf{B}), \quad \text{s.t. } \mathbf{B} \in \{\pm 1\}^{n \times r}, \ \mathbf{1}^{\top}\mathbf{B} = \mathbf{0}, \ \mathbf{B}^{\top}\mathbf{B} = n\mathbf{I}_r, \quad (1)$$

where $\mathbf{L}^o$ is the graph Laplacian based on the true affinity matrix $\mathbf{A}^{o1}$. The constraint $\mathbf{1}^{\top}\mathbf{B} = \mathbf{0}$ is imposed to maximize the information from each hash bit, which occurs when each bit leads to a *balanced* partitioning of the dataset $\mathcal{X}$. Another constraint $\mathbf{B}^{\top}\mathbf{B} = n\mathbf{I}_r$ makes $r$ bits mutually *uncorrelated* to minimize the redundancy among these bits. Problem (1) is NP-hard, and Weiss *et al.* [38] therefore solved a relaxed problem by dropping the discrete (binary) constraint $\mathbf{B} \in \{\pm 1\}^{n \times r}$ and making a simplifying assumption of data being distributed uniformly.

We leverage the anchor graph to replace $\mathbf{L}^o$ by the anchor graph Laplacian $\mathbf{L} = \mathbf{I}_n - \mathbf{A}$. Hence, the objective in Eq. (1) can be rewritten as a maximization problem:

$$\max_{\mathbf{B}} \quad \mathrm{tr}(\mathbf{B}^{\top}\mathbf{A}\mathbf{B}), \quad \text{s.t. } \mathbf{B} \in \{1, -1\}^{n \times r}, \ \mathbf{1}^{\top}\mathbf{B} = \mathbf{0}, \ \mathbf{B}^{\top}\mathbf{B} = n\mathbf{I}_r. \quad (2)$$

In [23], the solution to this problem is obtained via *spectral relaxation* [33] in which $\mathbf{B}$ is relaxed to be a matrix of reals followed by a thresholding step (threshold is 0) that brings the final discrete $\mathbf{B}$. Unfortunately, this procedure may result in poor codes due to amplification of the error caused by the relaxation as the code length $r$ increases. To this end, we propose to directly solve the binary codes $\mathbf{B}$ without resorting to such error-prone relaxations.

Let us define a set $\Omega = \left\{\mathbf{Y} \in \mathbb{R}^{n \times r} | \mathbf{1}^{\top}\mathbf{Y} = \mathbf{0}, \mathbf{Y}^{\top}\mathbf{Y} = n\mathbf{I}_r \right\}$. Then we formulate a more general graph hashing framework which softens the last two hard constraints in Eq. (2) as:

$$\max_{\mathbf{B}} \quad \mathrm{tr}(\mathbf{B}^{\top}\mathbf{A}\mathbf{B}) - \frac{\rho}{2}\mathrm{dist}^2(\mathbf{B}, \Omega), \quad \text{s.t. } \mathbf{B} \in \{1, -1\}^{n \times r}, \quad (3)$$

where $\mathrm{dist}(\mathbf{B}, \Omega) = \min_{\mathbf{Y} \in \Omega} \|\mathbf{B} - \mathbf{Y}\|_{\mathrm{F}}$ measures the distance from any matrix $\mathbf{B}$ to the set $\Omega$, and $\rho \geq 0$ is a tuning parameter. If problem (2) is feasible, we can enforce $\mathrm{dist}(\mathbf{B}, \Omega) = 0$ in Eq. (3) by imposing a very large $\rho$, thereby turning problem (3) into problem (2). However, in Eq. (3) we allow a certain discrepancy between $\mathbf{B}$ and $\Omega$ (controlled by $\rho$), which makes problem (3) more flexible. Since $\mathrm{tr}(\mathbf{B}^{\top}\mathbf{B}) = \mathrm{tr}(\mathbf{Y}^{\top}\mathbf{Y}) = nr$, problem (3) can be equivalently transformed to the following problem:

$$\max_{\mathbf{B}, \mathbf{Y}} \quad \mathcal{Q}(\mathbf{B}, \mathbf{Y}) := \mathrm{tr}(\mathbf{B}^{\top}\mathbf{A}\mathbf{B}) + \rho \mathrm{tr}(\mathbf{B}^{\top}\mathbf{Y}),$$

$$\text{s.t. } \mathbf{B} \in \{1, -1\}^{n \times r}, \ \mathbf{Y} \in \mathbb{R}^{n \times r}, \ \mathbf{1}^{\top}\mathbf{Y} = \mathbf{0}, \ \mathbf{Y}^{\top}\mathbf{Y} = n\mathbf{I}_r. \quad (4)$$

We call the code learning model formulated in Eq. (4) as *Discrete Graph Hashing* (DGH). Because concurrently imposing $\mathbf{B} \in \{\pm 1\}^{n \times r}$ and $\mathbf{B} \in \Omega$ will make graph hashing computationally intractable, DGH does not pursue the latter constraint but penalizes the distance from the target code matrix $\mathbf{B}$ to $\Omega$. Different from the previous graph hashing methods which discard the discrete constraint $\mathbf{B} \in \{\pm 1\}^{n \times r}$ to obtain continuously relaxed $\mathbf{B}$, our DGH model enforces this constraint to directly achieve discrete $\mathbf{B}$. As a result, DGH yields nearly balanced and uncorrelated binary bits. In Section 3, we will propose a computationally tractable optimization algorithm to solve this discrete programming problem in Eq. (4).

**Algorithm 1** Signed Gradient Method (SGM) for **B**-Subproblem

---

**Input:** $\mathbf{B}^{(0)} \in \{1, -1\}^{n \times r}$ and $\mathbf{Y} \in \Omega$.
$j := 0$; **repeat** $\mathbf{B}^{(j+1)} := \mathrm{sgn}\big(\mathcal{C}\big(2\mathbf{A}\mathbf{B}^{(j)} + \rho\mathbf{Y}, \mathbf{B}^{(j)}\big)\big)$, $j := j+1$, **until** $\mathbf{B}^{(j)}$ converges.
**Output:** $\mathbf{B} = \mathbf{B}^{(j)}$.

---

**Out-of-Sample Hashing.** Since a hashing scheme should be able to generate the hash code for any data point $\boldsymbol{q} \in \mathbb{R}^d$ beyond the points in the training set $\mathcal{X}$, here we address the out-of-sample extension of the DGH model. Similar to the objective in Eq. (1), we minimize the Hamming distances between a novel data point $\boldsymbol{q}$ and its neighbors (revealed by the affinity function $A$) in $\mathcal{X}$ as

$$\boldsymbol{b}(\boldsymbol{q}) \in \arg \min_{\boldsymbol{b}(\boldsymbol{q}) \in \{\pm 1\}^r} \frac{1}{2} \sum_{i=1}^{n} \left\| \boldsymbol{b}(\boldsymbol{q}) - \boldsymbol{b}_i^* \right\|^2 A(\boldsymbol{q}, \boldsymbol{x}_i) = \arg \max_{\boldsymbol{b}(\boldsymbol{q}) \in \{\pm 1\}^r} \left\langle \boldsymbol{b}(\boldsymbol{q}), (\mathbf{B}^*)^\top \mathbf{Z} \boldsymbol{\Lambda}^{-1} \boldsymbol{z}(\boldsymbol{q}) \right\rangle,$$

where $\mathbf{B}^* = [\boldsymbol{b}_1^*, \cdots, \boldsymbol{b}_n^*]^\top$ is the solution of problem (4). After pre-computing a matrix $\mathbf{W} = (\mathbf{B}^*)^\top \mathbf{Z} \boldsymbol{\Lambda}^{-1} \in \mathbb{R}^{r \times m}$ in the training phase, one can compute the hash code $\boldsymbol{b}^*(\boldsymbol{q}) = \mathrm{sgn}\big(\mathbf{W}\boldsymbol{z}(\boldsymbol{q})\big)$ for any novel data point $\boldsymbol{q}$ very efficiently.

## 3 Alternating Maximization

The graph hashing problem in Eq. (4) is essentially a nonlinear mixed-integer program involving both discrete variables in $\mathbf{B}$ and continuous variables in $\mathbf{Y}$. It turns out that problem (4) is generally NP-hard and also difficult to approximate. In specific, since the Max-Cut problem is a special case of problem (4) when $\rho = 0$ and $r = 1$, there exists no polynomial-time algorithm which can achieve the global optimum, or even an approximate solution with its objective value beyond 16/17 of the global maximum unless $\mathtt{P} = \mathtt{NP}$ [11]. To this end, we propose a tractable alternating maximization algorithm to optimize problem (4), leading to good hash codes which are demonstrated to exhibit superior search performance through extensive experiments conducted in Section 5.

The proposed algorithm proceeds by alternately solving the $\mathbf{B}$-subproblem

$$\max_{\mathbf{B} \in \{\pm 1\}^{n \times r}} f(\mathbf{B}) := \mathrm{tr}\big(\mathbf{B}^\top \mathbf{A}\mathbf{B}\big) + \rho\,\mathrm{tr}\big(\mathbf{Y}^\top \mathbf{B}\big) \tag{5}$$

and the $\mathbf{Y}$-subproblem

$$\max_{\mathbf{Y} \in \mathbb{R}^{n \times r}} \mathrm{tr}\big(\mathbf{B}^\top \mathbf{Y}\big), \quad \text{s.t. } \mathbf{1}^\top \mathbf{Y} = \mathbf{0}, \ \mathbf{Y}^\top \mathbf{Y} = n\mathbf{I}_r. \tag{6}$$

In what follows, we propose an iterative ascent procedure called *Signed Gradient Method* for subproblem (5) and derive a closed-form optimal solution to subproblem (6). As we can show, our alternating algorithm is provably convergent. Schemes for choosing good initializations are also discussed. Due to the space limit, all the proofs of lemmas, theorems and propositions presented in this section are placed in the supplemental material.

### 3.1 B-Subproblem

We tackle subproblem (5) with a simple iterative ascent procedure described in Algorithm 1. In the $j$-th iteration, we define a local function $\hat{f}_j(\mathbf{B})$ that linearizes $f(\mathbf{B})$ at the point $\mathbf{B}^{(j)}$, and employ $\hat{f}_j(\mathbf{B})$ as a surrogate of $f(\mathbf{B})$ for discrete optimization. Given $\mathbf{B}^{(j)}$, the next discrete point is derived as $\mathbf{B}^{(j+1)} \in \arg\max_{\mathbf{B} \in \{\pm 1\}^{n \times r}} \hat{f}_j(\mathbf{B}) := f\big(\mathbf{B}^{(j)}\big) + \big\langle \nabla f\big(\mathbf{B}^{(j)}\big), \mathbf{B} - \mathbf{B}^{(j)} \big\rangle$. Note that since $\nabla f\big(\mathbf{B}^{(j)}\big)$ may include zero entries, multiple solutions for $\mathbf{B}^{(j+1)}$ could exist. To avoid this ambiguity, we introduce the function $\mathcal{C}(x, y) = \begin{cases} x, x \neq 0 \\ y, x = 0 \end{cases}$ to specify the following update:

$$\mathbf{B}^{(j+1)} := \mathrm{sgn}\Big(\mathcal{C}\big(\nabla f\big(\mathbf{B}^{(j)}\big), \mathbf{B}^{(j)}\big)\Big) = \mathrm{sgn}\Big(\mathcal{C}\big(2\mathbf{A}\mathbf{B}^{(j)} + \rho\mathbf{Y}, \mathbf{B}^{(j)}\big)\Big), \tag{7}$$

in which $\mathcal{C}$ is applied in an element-wise manner, and no update is carried out to the entries where $\nabla f\big(\mathbf{B}^{(j)}\big)$ vanishes.

Due to the PSD property of the matrix $\mathbf{A}$, $f$ is a convex function and thus $f(\mathbf{B}) \geq \hat{f}_j(\mathbf{B})$ for any $\mathbf{B}$. Taking advantage of the fact $f\big(\mathbf{B}^{(j+1)}\big) \geq \hat{f}_j\big(\mathbf{B}^{(j+1)}\big) \geq \hat{f}_j\big(\mathbf{B}^{(j)}\big) \equiv f\big(\mathbf{B}^{(j)}\big)$, Lemma 1 ensures that both the sequence of cost values $\big\{f(\mathbf{B}^{(j)})\big\}$ and the sequence of iterates $\big\{\mathbf{B}^{(j)}\big\}$ converge.

---
**Algorithm 2** Discrete Graph Hashing (DGH)
---
 **Input:** $\mathbf{B}_0 \in \{1, -1\}^{n \times r}$ and $\mathbf{Y}_0 \in \Omega$.
 $k := 0$;
 **repeat** $\mathbf{B}_{k+1} := \mathsf{SGM}(\mathbf{B}_k, \mathbf{Y}_k)$, $\mathbf{Y}_{k+1} \in \Phi(\mathbf{J}\mathbf{B}_{k+1})$, $k := k+1$, **until** $\mathcal{Q}(\mathbf{B}_k, \mathbf{Y}_k)$ converges.
 **Output:** $\mathbf{B}^* = \mathbf{B}_k, \mathbf{Y}^* = \mathbf{Y}_k$.
---

**Lemma 1.** *If $\{\mathbf{B}^{(j)}\}$ is the sequence of iterates produced by Algorithm 1, then $f(\mathbf{B}^{(j+1)}) \geq f(\mathbf{B}^{(j)})$ holds for any integer $j \geq 0$, and both $\{f(\mathbf{B}^{(j)})\}$ and $\{\mathbf{B}^{(j)}\}$ converge.*

Our idea of optimizing a proxy function $\hat{f}_j(\mathbf{B})$ can be considered as a special case of majorization methodology exploited in the field of optimization. The majorization method typically deals with a generic constrained optimization problem: $\min g(\boldsymbol{x})$, s.t. $\boldsymbol{x} \in \mathcal{F}$, where $g : \mathbb{R}^n \mapsto \mathbb{R}$ is a continuous function and $\mathcal{F} \subseteq \mathbb{R}^n$ is a compact set. The majorization method starts with a feasible point $\boldsymbol{x}_0 \in \mathcal{F}$, and then proceeds by setting $\boldsymbol{x}_{j+1}$ as a minimizer of $\hat{g}_j(\boldsymbol{x})$ over $\mathcal{F}$, where $\hat{g}_j$ satisfying $\hat{g}_j(\boldsymbol{x}_j) = g(\boldsymbol{x}_j)$ and $\hat{g}_j(\boldsymbol{x}) \geq g(\boldsymbol{x}) \, \forall \boldsymbol{x} \in \mathcal{F}$ is called a majorization function of $g$ at $\boldsymbol{x}_j$. In specific, in our scenario, problem (5) is equivalent to $\min_{\mathbf{B} \in \{\pm 1\}^{n \times r}} -f(\mathbf{B})$, and the linear surrogate $-\hat{f}_j$ is a majorization function of $-f$ at point $\mathbf{B}^{(j)}$. The majorization method was first systematically introduced by [5] to deal with multidimensional scaling problems, although the EM algorithm [7], proposed at the same time, also falls into the framework of majorization methodology. Since then, the majorization method has played an important role in various statistics problems such as multi-dimensional data analysis [12], hyperparameter learning [8], conditional random fields and latent likelihoods [13], and so on.

### 3.2 Y-Subproblem

An analytical solution to subproblem (6) can be obtained with the aid of a centering matrix $\mathbf{J} = \mathbf{I}_n - \frac{1}{n}\mathbf{1}\mathbf{1}^\top$. Write the singular value decomposition (SVD) of $\mathbf{J}\mathbf{B}$ as $\mathbf{J}\mathbf{B} = \mathbf{U}\boldsymbol{\Sigma}\mathbf{V}^\top = \sum_{k=1}^{r'} \sigma_k \boldsymbol{u}_k \boldsymbol{v}_k^\top$, where $r' \leq r$ is the rank of $\mathbf{J}\mathbf{B}$, $\sigma_1, \cdots, \sigma_{r'}$ are the positive singular values, and $\mathbf{U} = [\boldsymbol{u}_1, \cdots, \boldsymbol{u}_{r'}]$ and $\mathbf{V} = [\boldsymbol{v}_1, \cdots, \boldsymbol{v}_{r'}]$ contain the left- and right-singular vectors, respectively. Then, by employing a Gram-Schmidt process, one can easily construct matrices $\bar{\mathbf{U}} \in \mathbb{R}^{n \times (r-r')}$ and $\bar{\mathbf{V}} \in \mathbb{R}^{r \times (r-r')}$ such that $\bar{\mathbf{U}}^\top \bar{\mathbf{U}} = \mathbf{I}_{r-r'}$, $[\mathbf{U}\ \mathbf{1}]^\top \bar{\mathbf{U}} = \mathbf{0}$, and $\bar{\mathbf{V}}^\top \bar{\mathbf{V}} = \mathbf{I}_{r-r'}$, $\mathbf{V}^\top \bar{\mathbf{V}} = \mathbf{0}^2$. Now we are ready to characterize a closed-form solution of the $\mathbf{Y}$-subproblem by Lemma 2.

**Lemma 2.** $\mathbf{Y}^\star = \sqrt{n}[\mathbf{U}\ \bar{\mathbf{U}}][\mathbf{V}\ \bar{\mathbf{V}}]^\top$ *is an optimal solution to the $\mathbf{Y}$-subproblem in Eq. (6).*

For notational convenience, we define the set of all matrices in the form of $\sqrt{n}[\mathbf{U}\ \bar{\mathbf{U}}][\mathbf{V}\ \bar{\mathbf{V}}]^\top$ as $\Phi(\mathbf{J}\mathbf{B})$. Lemma 2 reveals that any matrix in $\Phi(\mathbf{J}\mathbf{B})$ is an optimal solution to subproblem (6). In practice, to compute such an optimal $\mathbf{Y}^\star$, we perform the eigendecomposition over the small $r \times r$ matrix $\mathbf{B}^\top \mathbf{J}\mathbf{B}$ to have $\mathbf{B}^\top \mathbf{J}\mathbf{B} = [\mathbf{V}\ \bar{\mathbf{V}}] \begin{bmatrix} \boldsymbol{\Sigma}^2 & \mathbf{0} \\ \mathbf{0} & \mathbf{0} \end{bmatrix} [\mathbf{V}\ \bar{\mathbf{V}}]^\top$, which gives $\mathbf{V}, \bar{\mathbf{V}}, \boldsymbol{\Sigma}$, and immediately leads to $\mathbf{U} = \mathbf{J}\mathbf{B}\mathbf{V}\boldsymbol{\Sigma}^{-1}$. The matrix $\bar{\mathbf{U}}$ is initially set to a random matrix followed by the aforementioned Gram-Schmidt orthogonalization. It can be seen that $\mathbf{Y}^\star$ is uniquely optimal when $r' = r$ (*i.e.*, $\mathbf{J}\mathbf{B}$ is full column rank).

### 3.3 DGH Algorithm

The proposed alternating maximization algorithm, also referred to as Discrete Graph Hashing (DGH), for solving the raw problem in Eq. (4) is summarized in Algorithm 2, in which we introduce $\mathsf{SGM}(\cdot, \cdot)$ to represent the functionality of Algorithm 1. The convergence of Algorithm 2 is guaranteed by Theorem 1, whose proof is based on the nature of the proposed alternating maximization procedure that always generates a monotonically non-decreasing and bounded sequence.

**Theorem 1.** *If $\{(\mathbf{B}_k, \mathbf{Y}_k)\}$ is the sequence generated by Algorithm 2, then $\mathcal{Q}(\mathbf{B}_{k+1}, \mathbf{Y}_{k+1}) \geq \mathcal{Q}(\mathbf{B}_k, \mathbf{Y}_k)$ holds for any integer $k \geq 0$, and $\{\mathcal{Q}(\mathbf{B}_k, \mathbf{Y}_k)\}$ converges starting with any feasible initial point $(\mathbf{B}_0, \mathbf{Y}_0)$.*

**Initialization.** Since the DGH algorithm deals with discrete and non-convex optimization, a good choice of an initial point $(\mathbf{B}_0, \mathbf{Y}_0)$ is vital. Here we suggest two different initial points which are both feasible to problem (4).

Let us perform the eigendecomposition over $\mathbf{A}$ to obtain $\mathbf{A} = \mathbf{P}\mathbf{\Theta}\mathbf{P}^\top = \sum_{k=1}^m \theta_k \boldsymbol{p}_k \boldsymbol{p}_k^\top$, where $\theta_1, \cdots, \theta_m$ are the eigenvalues arranged in a non-increasing order, and $\boldsymbol{p}_1, \cdots, \boldsymbol{p}_m$ are the corresponding normalized eigenvectors. We write $\mathbf{\Theta} = \mathrm{diag}(\theta_1, \cdots, \theta_m)$ and $\mathbf{P} = [\boldsymbol{p}_1, \cdots, \boldsymbol{p}_m]$. Note that $\theta_1 = 1$ and $\boldsymbol{p}_1 = 1/\sqrt{n}$. The first initialization used is $\big(\mathbf{Y}_0 = \sqrt{n}\mathbf{H}, \mathbf{B}_0 = \mathrm{sgn}(\mathbf{H})\big)$, where $\mathbf{H} = [\boldsymbol{p}_2, \cdots, \boldsymbol{p}_{r+1}] \in \mathbb{R}^{n \times r}$. The initial codes $\mathbf{B}_0$ were used as the final codes by [23].

Alternatively, $\mathbf{Y}_0$ can be allowed to consist of orthonormal columns within the column space of $\mathbf{H}$, i.e., $\mathbf{Y}_0 = \sqrt{n}\mathbf{H}\mathbf{R}$ subject to some orthogonal matrix $\mathbf{R} \in \mathbb{R}^{r \times r}$. We can obtain $\mathbf{R}$ along with $\mathbf{B}_0$ by solving a new discrete optimization problem:

$$\max_{\mathbf{R}, \mathbf{B}_0} \ \mathrm{tr}\big(\mathbf{R}^\top \mathbf{H}^\top \mathbf{A} \mathbf{B}_0\big), \quad \text{s.t. } \mathbf{R} \in \mathbb{R}^{r \times r}, \ \mathbf{R}\mathbf{R}^\top = \mathbf{I}_r, \ \mathbf{B}_0 \in \{1, -1\}^{n \times r}, \tag{8}$$

which is motivated by the proposition below.

**Proposition 1.** *For any orthogonal matrix $\mathbf{R} \in \mathbb{R}^{r \times r}$ and any binary matrix $\mathbf{B} \in \{1, -1\}^{n \times r}$, we have $\mathrm{tr}\big(\mathbf{B}^\top \mathbf{A} \mathbf{B}\big) \geq \dfrac{1}{r}\mathrm{tr}^2\big(\mathbf{R}^\top \mathbf{H}^\top \mathbf{A} \mathbf{B}\big)$.*

Proposition 1 implies that the optimization in Eq. (8) can be interpreted as to maximize a lower bound of $\mathrm{tr}\big(\mathbf{B}^\top \mathbf{A} \mathbf{B}\big)$ which is the first term of the objective $\mathcal{Q}(\mathbf{B}, \mathbf{Y})$ in the original problem (4). We still exploit an alternating maximization procedure to solve problem (8). Noticing $\mathbf{A}\mathbf{H} = \mathbf{H}\hat{\mathbf{\Theta}}$ where $\hat{\mathbf{\Theta}} = \mathrm{diag}(\theta_2, \cdots, \theta_{r+1})$, the objective in Eq. (8) is equal to $\mathrm{tr}\big(\mathbf{R}^\top \hat{\mathbf{\Theta}} \mathbf{H}^\top \mathbf{B}_0\big)$. The alternating procedure starts with $\mathbf{R}^0 = \mathbf{I}_r$, and then makes the simple updates $\mathbf{B}_0^j := \mathrm{sgn}\big(\mathbf{H}\hat{\mathbf{\Theta}}\mathbf{R}^j\big)$, $\mathbf{R}^{j+1} := \tilde{\mathbf{U}}_j \tilde{\mathbf{V}}_j^\top$ for $j = 0, 1, 2, \cdots$, where $\tilde{\mathbf{U}}_j, \tilde{\mathbf{V}}_j \in \mathbb{R}^{r \times r}$ stem from the full SVD $\tilde{\mathbf{U}}_j \tilde{\mathbf{\Sigma}}_j \tilde{\mathbf{V}}_j^\top$ of the matrix $\hat{\mathbf{\Theta}}\mathbf{H}^\top \mathbf{B}_0^j$. When convergence is reached, we obtain the optimized rotation $\mathbf{R}$ that yields the second initialization $\big(\mathbf{Y}_0 = \sqrt{n}\mathbf{H}\mathbf{R}, \mathbf{B}_0 = \mathrm{sgn}(\mathbf{H}\hat{\mathbf{\Theta}}\mathbf{R})\big)$.

Empirically, we find that the second initialization typically gives a better objective value $\mathcal{Q}(\mathbf{B}_0, \mathbf{Y}_0)$ at the start than the first one, as it aims to maximize the lower bound of the first term in the objective $\mathcal{Q}$. We also observe that the second initialization often results in a higher objective value $\mathcal{Q}(\mathbf{B}^*, \mathbf{Y}^*)$ at convergence (Figs. 1-2 in the supplemental material show convergence curves of $\mathcal{Q}$ starting from the two initial points). We call DGH using the first and second initializations as DGH-I and DGH-R, respectively. Regarding the convergence property, we would like to point out that since the DGH algorithm (Algorithm 2) works on a mixed-integer objective, it is hard to quantify the convergence to a local optimum of the objective function $\mathcal{Q}$. Nevertheless, this does not affect the performance of our algorithm in practice. In our experiments in Section 5, we consistently find a convergent sequence $\{(\mathbf{B}_k, \mathbf{Y}_k)\}$ arriving at a good objective value when started with the suggested initializations.

## 4 Discussions

Here we analyze space and time complexities of DGH-I/DGH-R. The space complexity is $O\big((d + s + r)n\big)$ in the training stage and $O(rn)$ for storing hash codes in the test stage for DGH-I/DGH-R. Let $T_B$ and $T_G$ be the budget iteration numbers of optimizing the $\mathbf{B}$-subproblem and the whole DGH problem, respectively. Then, the training time complexity of DGH-I is $O\big(dmn + m^2 n + (mT_B + sT_B + r)rT_G n\big)$, and the training time complexity of DGH-R is $O\big(dmn + m^2 n + (mT_B + sT_B + r)rT_G n + r^2 T_R n\big)$, where $T_R$ is the budget iteration number for seeking the initial point via Eq. (8). Note that the time for finding anchors and building the anchor graph is $O(dmn)$ which is included in the above training time. Their test time (referring to encoding a query to an $r$-bit code) is both $O(dm + sr)$. In our experiments, we fix $m, s, T_B, T_G, T_R$ to constants independent of the dataset size $n$, and make $r \leq 128$. Thus, DGH-I/DGH-R enjoy linear training time and constant test time. It is worth mentioning again that the low-rank PSD property of the anchor graph affinity matrix $\mathbf{A}$ is advantageous for training DGH, permitting efficient matrix computations in $O(n)$ time, such as the eigendecomposition of $\mathbf{A}$ (encountered in initializations) and multiplying $\mathbf{A}$ with $\mathbf{B}$ (encountered in solving the $\mathbf{B}$-subproblem with Algorithm 1).

It is interesting to point out that DGH falls into the asymmetric hashing category [26] in the sense that hash codes are generated differently for samples within the dataset and queries outside the dataset. Unlike most existing hashing techniques, DGH directly solves the hash codes $\mathbf{B}^*$ of the training samples via the proposed discrete optimization in Eq. (4) without relying on any explicit or predefined hash functions. On the other hand, the hash code for any query $\boldsymbol{q}$ is induced from the solved codes $\mathbf{B}^*$, leading to a hash function $\boldsymbol{b}^*(\boldsymbol{q}) = \mathrm{sgn}\big(\mathbf{W}\boldsymbol{z}(\boldsymbol{q})\big)$ parameterized by the matrix

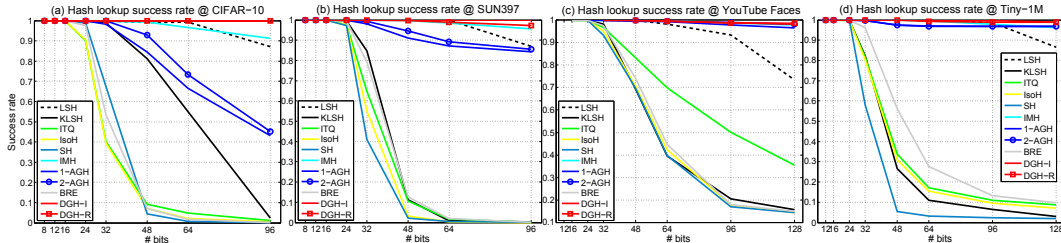

Figure 1: Hash lookup success rates for different hashing techniques. DGH tends to achieve nearly 100% success rates even for longer code lengths.

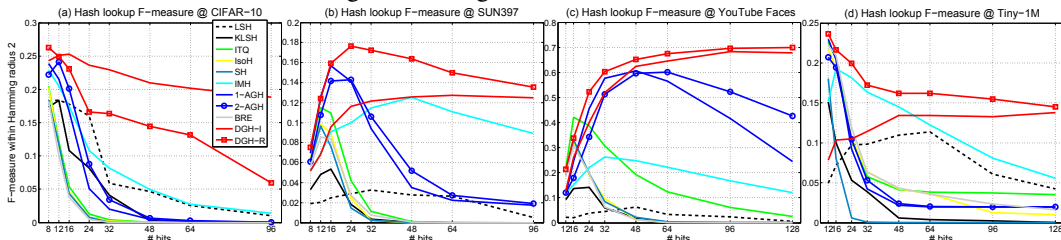

Figure 2: Mean F-measures of hash lookup within Hamming radius 2 for different techniques. DGH tends to retain good recall even for longer codes, leading to much higher F-measures than the others.

$\mathbf{W}$ which was computed using $\mathbf{B}^*$. While the hashing mechanisms for producing $\mathbf{B}^*$ and $\boldsymbol{b}^*(\boldsymbol{q})$ are distinct, they are tightly coupled and prone to be adaptive to specific datasets. The flexibility of the asymmetric hashing nature of DGH is validated through the experiments shown in the next section.

## 5 Experiments

We conduct large-scale similarity search experiments on four benchmark datasets: **CIFAR-10** [15], **SUN397** [40], **YouTube Faces** [39], and **Tiny-1M**. **CIFAR-10** is a labeled subset of the 80 Million Tiny Images dataset [35], which consists of 60K images from ten object categories with each image represented by a 512-dimensional GIST feature vector [29]. **SUN397** contains about 108K images from 397 scene categories, where each image is represented by a 1,600-dimensional feature vector extracted by PCA from 12,288-dimensional Deep Convolutional Activation Features [10]. The raw **YouTube Faces** dataset contains 1,595 different people, from which we choose 340 people such that each one has at least 500 images to form a subset of 370,319 face images, and represent each face image as a 1,770-dimensional LBP feature vector [1]. **Tiny-1M** is one million subset of the 80M tiny images, where each image is represented by a 384-dimensional GIST vector. In **CIFAR-10**, 100 images are sampled uniformly randomly from each object category to form a separate test (query) set of 1K images; in **SUN397**, 100 images are sampled uniformly randomly from each of the 18 largest scene categories to form a test set of 1.8K images; in **YouTube Faces**, the test set includes 3.8K face images which are evenly sampled from the 38 people each containing more than 2K faces; in **Tiny-1M**, a separate subset of 5K images randomly sampled from the 80M images is used as the test set. In the first three datasets, groundtruth neighbors are defined based on whether two samples share the same class label; in **Tiny-1M** which does not have full annotations, we define groundtruth neighbors for a given query as the samples among the top 2% $\ell_2$ distances from the query in the 1M training set, so each query has 20K groundtruth neighbors.

We evaluate twelve unsupervised hashing methods including: two randomized methods LSH [2] and Kernelized LSH (KLSH) [17], two linear projection based methods Iterative Quantization (ITQ) [9] and Isotropic Hashing (IsoH) [14], two spectral methods Spectral Hashing (SH) [38] and its weighted version MDSH [37], one manifold based method Inductive Manifold Hashing (IMH) [32], two existing graph-based methods One-Layer Anchor Graph Hashing (1-AGH) and Two-Layer Anchor Graph Hashing (2-AGH) [23], one distance preservation method Binary Reconstruction Embeddings (BRE) [16] (unsupervised version), and our proposed discrete optimization based methods DGH-I and DGH-R. We use the publicly available codes of the competing methods, and follow the conventional parameter settings therein. In particular, we use the Gaussian kernel and 300 randomly sampled exemplars (anchors) to run KLSH; IMH, 1-AGH, 2-AGH, DGH-I and DGH-R also use $m = 300$ anchors (obtained by K-means clustering with 5 iterations) for fair comparison. This choice of $m$ gives a good trade-off between hashing speed and performance. For 1-AGH, 2-AGH, DGH-I and DGH-R that all use anchor graphs, we adopt the same construction parameters $s, t$ on each dataset ($s = 3$ and $t$ is tuned following AGH), and $\ell_2$ distance as $\mathcal{D}(\cdot)$. For BRE, we uniformly

Table 1: Hamming ranking performance on **YouTube Faces** and **Tiny-1M**. $r$ denotes the number of hash bits used in the hashing methods. All training and test times are in seconds.

| Method | YouTube Faces | | | | | Tiny-1M | | | | |
|---|---|---|---|---|---|---|---|---|---|---|
| | Mean Precision / Top-2K | | | TrainTime | TestTime | Mean Precision / Top-20K | | | TrainTime | TestTime |
| | $r=48$ | $r=96$ | $r=128$ | $r=128$ | $r=128$ | $r=48$ | $r=96$ | $r=128$ | $r=128$ | $r=128$ |
| $\ell_2$ Scan | 0.7591 | | | – | | 1 | | | – | |
| LSH | 0.0830 | 0.1005 | 0.1061 | 6.4 | $1.8\times10^{-5}$ | 0.1155 | 0.1324 | 0.1766 | 6.1 | $1.0\times10^{-5}$ |
| KLSH | 0.3982 | 0.5210 | 0.5871 | 16.1 | $4.8\times10^{-5}$ | 0.3054 | 0.4105 | 0.4705 | 20.7 | $4.6\times10^{-5}$ |
| ITQ | 0.7017 | 0.7493 | 0.7562 | 169.0 | $1.8\times10^{-5}$ | 0.3925 | 0.4726 | 0.5052 | 297.3 | $1.0\times10^{-5}$ |
| IsoH | 0.6093 | 0.6962 | 0.7058 | 73.6 | $1.8\times10^{-5}$ | 0.3896 | 0.4816 | 0.5161 | 13.5 | $1.0\times10^{-5}$ |
| SH | 0.5897 | 0.6655 | 0.6736 | 108.9 | $2.0\times10^{-4}$ | 0.1857 | 0.1923 | 0.2079 | 61.4 | $1.6\times10^{-4}$ |
| MDSH | 0.6110 | 0.6752 | 0.6795 | 118.8 | $4.9\times10^{-5}$ | 0.3312 | 0.3878 | 0.3955 | 193.6 | $2.8\times10^{-5}$ |
| IMH | 0.3150 | 0.3641 | 0.3889 | 92.1 | $2.3\times10^{-5}$ | 0.2257 | 0.2497 | 0.2557 | 139.3 | $2.7\times10^{-5}$ |
| 1-AGH | 0.7138 | 0.7571 | 0.7646 | 84.1 | $2.1\times10^{-5}$ | 0.4061 | 0.4117 | 0.4107 | 141.4 | $3.4\times10^{-5}$ |
| 2-AGH | 0.6727 | 0.7377 | 0.7521 | 94.7 | $3.5\times10^{-5}$ | 0.3925 | 0.4099 | 0.4152 | 272.5 | $4.7\times10^{-5}$ |
| BRE | 0.5564 | 0.6238 | 0.6483 | 10372.0 | $9.0\times10^{-5}$ | 0.3943 | 0.4836 | 0.5218 | 8419.0 | $8.8\times10^{-5}$ |
| **DGH-I** | 0.7086 | 0.7644 | 0.7750 | 402.6 | $2.1\times10^{-5}$ | 0.4045 | 0.4865 | 0.5178 | 1769.4 | $3.3\times10^{-5}$ |
| **DGH-R** | **0.7245** | **0.7672** | **0.7805** | 408.9 | $2.1\times10^{-5}$ | **0.4208** | **0.5006** | **0.5358** | 2793.4 | $3.3\times10^{-5}$ |

randomly sample 1K, and 2K training samples to train the distance preservations on **CIFAR-10** & **SUN397**, and **YouTube Faces** & **Tiny-1M**, respectively. For DGH-I and DGH-R, we set the penalty parameter $\rho$ to the same value in $[0.1, 5]$ on each dataset, and fix $T_R = 100$, $T_B = 300$, $T_G = 20$.

We employ two widely used search procedures *hash lookup* and *Hamming ranking* with 8 to 128 hash bits for evaluations. The Hamming ranking procedure ranks the dataset samples according to their Hamming distances to a given query, while the hash lookup procedure finds all the points within a certain Hamming radius away from the query. Since hash lookup can be achieved in constant time by using a single hash table, it is the main focus of this work. We carry out hash lookup within a Hamming ball of radius 2 centered on each query, and report the search recall and F-measure which are averaged over all queries for each dataset. Note that if table lookup fails to find any neighbors within a given radius for a query, we call it a failed query and assign it zero recall and F-measure. To quantify the failed queries, we report the hash lookup success rate which gives the proportion of the queries for which at least one neighbor is retrieved. For Hamming ranking, mean average precision (MAP) and mean precision of top-retrieved samples are computed.

The hash lookup results are shown in Figs. 1-2. DGH-I/DGH-R achieve the highest (close to 100%) hash lookup success rates, and DGH-I is slightly better than DGH-R. The reason is that the asymmetric hashing scheme exploited by DGH-I/DGH-R poses a tight linkage to connect queries and database samples, providing a more adaptive out-of-sample extension than the traditional symmetric hashing schemes used by the competing methods. Also, DGH-R achieves the highest F-measure except on **CIFAR-10**, where DGH-I is highest while DGH-R is the second. The F-measures of KLSH, IsoH, SH and BRE deteriorate quickly and are with very poor values ($< 0.05$) when $r \geq 48$ due to poor recall[3]. Although IMH achieves nice hash lookup succuss rates, its F-measures are much lower than DGH-I/DGH-R due to lower precision. MDSH produces the same hash bits as SH, so is not included in the hash lookup experiments. DGH-I/DGH-R employ the proposed discrete optimization to yield high-quality codes that preserve the local neighborhood of each data point within a small Hamming ball, so obtain much higher search accuracy in F-measure and recall than SH, 1-AGH and 2-AGH which rely on relaxed optimizations and degrade drastically when $r \geq 48$. Finally, we report the Hamming ranking results in Table 1 and the table in the sup-material, which clearly show the superiority of DGH-R over the competing methods in MAP and mean precision; on the first three datasets, DGH-R even outperforms exhaustive $\ell_2$ scan. The training time of DGH-I/DGH-R is acceptable and faster than BRE, and their test time (*i.e.*, coding time since hash lookup time is small enough to be ignored) is comparable with 1-AGH.

## 6 Conclusion

This paper investigated a pervasive problem of not enforcing the discrete constraints in optimization pertaining to most existing hashing methods. Instead of resorting to error-prone continuous relaxations, we introduced a novel discrete optimization technique that learns the binary hash codes directly. To achieve this, we proposed a tractable alternating maximization algorithm which solves two interesting subproblems and provably converges. When working with a neighborhood graph, the proposed method yields high-quality codes to well preserve the neighborhood structure inherent in the data. Extensive experimental results on four large datasets up to one million showed that our discrete optimization based graph hashing technique is highly competitive.

## Footnotes

[1] The spectral hashing method in [38] did not compute the true affinity matrix $\mathbf{A}^o$ because of the scalability issue, but instead used a complete graph built over 1D PCA embeddings.

[2]Note that when $r' = r$, $\bar{\mathbf{U}}$ and $\bar{\mathbf{V}}$ are nothing but $\mathbf{0}$.

[3]The recall results are shown in Fig. 3 of the supplemental material, which indicate that DGH-I achieves the highest recall except on **YouTube Faces**, where DGH-R is highest while DGH-I is the second.

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
