[Supplementary Material · NIPS14_dgh_sup.pdf]

# Supplemental Material for "Discrete Graph Hashing"

**Wei Liu**[†]  **Cun Mu**[‡]  **Sanjiv Kumar**[♯]  **Shih-Fu Chang**[‡]

[†]IBM T. J. Watson Research Center  [‡]Columbia University  [♯]Google Research

weiliu@us.ibm.com  cm3052@columbia.edu
sfchang@ee.columbia.edu  sanjivk@google.com

## 1 Proofs

**Lemma 1.** *If* $\{\mathbf{B}^{(j)}\}$ *is the sequence of iterates produced by Algorithm 1, then* $f(\mathbf{B}^{(j+1)}) \geq f(\mathbf{B}^{(j)})$ *holds for any integer* $j \geq 0$, *and both* $\{f(\mathbf{B}^{(j)})\}$ *and* $\{\mathbf{B}^{(j)}\}$ *converge.*

*Proof.* Since the $\mathbf{B}$-subproblem in Eq. (5) of the main paper is to maximize a continuous function over a compact (*i.e.*, closed and bounded) set, its optimal objective function value $f^*$ is finite. As $\mathbf{B}^{(j)}$ is feasible for each $j$, the sequence $\{f(\mathbf{B}^{(j)})\}$ is bounded from above.

By definition, $\hat{f}_j(\mathbf{B}^{(j)}) \equiv f(\mathbf{B}^{(j)})$. Since $\mathbf{B}^{(j+1)} \in \arg\max_{\mathbf{B} \in \{\pm 1\}^{n \times r}} \hat{f}_j(\mathbf{B})$, we have $\hat{f}_j(\mathbf{B}^{(j+1)}) \geq \hat{f}_j(\mathbf{B}^{(j)})$. Also, as $\mathbf{A}$ is positive semidefinite, $f(\mathbf{B})$ is a convex function. Therefore, $f(\mathbf{B}) \geq \hat{f}_j(\mathbf{B})$ at any $\mathbf{B} \in \{1, -1\}^{n \times r}$, which implies that $f(\mathbf{B}^{(j+1)}) \geq \hat{f}_j(\mathbf{B}^{(j+1)})$. Putting all the above together, we have

$$f(\mathbf{B}^{(j+1)}) \geq \hat{f}_j(\mathbf{B}^{(j+1)}) \geq \hat{f}_j(\mathbf{B}^{(j)}) = f(\mathbf{B}^{(j)}), \ \forall j \in \mathbb{Z}, \ j \geq 0.$$

Together with the fact that $\{f(\mathbf{B}^{(j)})\}$ is bounded from above, it turns out that the monotonically non-decreasing sequence $\{f(\mathbf{B}^{(j)})\}$ converges.

The convergence of $\{\mathbf{B}^{(j)}\}$ is established based on the following three key facts.

First, if $\mathbf{B}^{(j+1)} = \mathbf{B}^{(j)}$, then due to the update in Eq. (7) of the main paper, we have $\mathbf{B}^{(j')} \equiv \mathbf{B}^{(j)}$ for any integer $j' \geq j + 1$.

Second, if $\mathbf{B}^{(j+1)} \neq \mathbf{B}^{(j)}$, then we can infer that the entries of the gradient $\nabla f(\mathbf{B}^{(j)})$ are not all zeros, and that there exists at least an entry $(i, k)$ $(1 \leq i \leq n, 1 \leq k \leq r)$ such that $\nabla_{i,k} f(\mathbf{B}^{(j)}) \neq 0$ and $(\mathbf{B}^{(j+1)})_{i,k} \neq (\mathbf{B}^{(j)})_{i,k}$. Due to the update in Eq. (7), at such an entry $(i,k)$, $(\mathbf{B}^{(j+1)})_{i,k} = \text{sgn}(\nabla_{i,k} f(\mathbf{B}^{(j)}))$, which implies that $\nabla_{i,k} f(\mathbf{B}^{(j)}) ((\mathbf{B}^{(j+1)})_{i,k} - (\mathbf{B}^{(j)})_{i,k}) > 0$. Then, we can derive as follows:

$$
\begin{aligned}
\hat{f}_j(\mathbf{B}^{(j+1)}) - \hat{f}_j(\mathbf{B}^{(j)}) &= \left\langle \nabla f(\mathbf{B}^{(j)}), \mathbf{B}^{(j+1)} - \mathbf{B}^{(j)} \right\rangle \\
&= \sum_{\substack{i,k \\ \nabla_{i,k} f(\mathbf{B}^{(j)}) = 0}} \nabla_{i,k} f(\mathbf{B}^{(j)}) \left( (\mathbf{B}^{(j+1)})_{i,k} - (\mathbf{B}^{(j)})_{i,k} \right) + \\
&\quad \sum_{\substack{i,k \\ \nabla_{i,k} f(\mathbf{B}^{(j)}) \neq 0 \\ (\mathbf{B}^{(j+1)})_{i,k} = (\mathbf{B}^{(j)})_{i,k}}} \nabla_{i,k} f(\mathbf{B}^{(j)}) \left( (\mathbf{B}^{(j+1)})_{i,k} - (\mathbf{B}^{(j)})_{i,k} \right) +
\end{aligned}
$$

$$\sum_{\substack{i,\,k \\ \nabla_{i,k} f(\mathbf{B}^{(j)}) \neq 0 \\ (\mathbf{B}^{(j+1)})_{i,k} \neq (\mathbf{B}^{(j)})_{i,k}}} \nabla_{i,k} f(\mathbf{B}^{(j)}) \left( (\mathbf{B}^{(j+1)})_{i,k} - (\mathbf{B}^{(j)})_{i,k} \right)$$

$$= 0 + 0 + \sum_{\substack{i,\,k \\ \nabla_{i,k} f(\mathbf{B}^{(j)}) \neq 0 \\ (\mathbf{B}^{(j+1)})_{i,k} \neq (\mathbf{B}^{(j)})_{i,k}}} \nabla_{i,k} f(\mathbf{B}^{(j)}) \left( (\mathbf{B}^{(j+1)})_{i,k} - (\mathbf{B}^{(j)})_{i,k} \right)$$

$$> 0.$$

As such, we have that once $\mathbf{B}^{(j+1)} \neq \mathbf{B}^{(j)}$,

$$f(\mathbf{B}^{(j+1)}) \geq \hat{f}_j(\mathbf{B}^{(j+1)}) > \hat{f}_j(\mathbf{B}^{(j)}) = f(\mathbf{B}^{(j)}),$$

which hence causes a strict increase from $f(\mathbf{B}^{(j)})$ to $f(\mathbf{B}^{(j+1)})$.

Third, there are only finite $f$ values in the cost sequence $\{f(\mathbf{B}^{(j)})\}$ because of finite feasible points in the set $\{1, -1\}^{n \times r}$.

Taking together the second and third facts, we find that there exists an integer $j \geq 0$ such that $\mathbf{B}^{(j+1)} = \mathbf{B}^{(j)}$, or otherwise infinite values will appear in $\{f(\mathbf{B}^{(j)})\}$. Incorporating the first fact, it can be easily arrived that there exists an integer $j \geq 0$ such that $\mathbf{B}^{(j')} \equiv \mathbf{B}^{(j)}$ for any integer $j' \geq j + 1$, which immediately indicates the convergence of $\{\mathbf{B}^{(j)}\}$. $\quad\square$

**Lemma 2.** $\mathbf{Y}^\star = \sqrt{n} [\mathbf{U} \ \bar{\mathbf{U}}][\mathbf{V} \ \bar{\mathbf{V}}]^\top$ *is an optimal solution to the* $\mathbf{Y}$*-subproblem in Eq. (6).*

*Proof.* We first prove that $\mathbf{Y}^\star$ is feasible to Eq. (6) in the main paper, *i.e.*, $\mathbf{Y}^\star \in \Omega$. Note that $\mathbf{1}^\top \mathbf{J} = \mathbf{0}$ and hence $\mathbf{1}^\top \mathbf{JB} = \mathbf{0}$. Because $\mathbf{JB}$ and $\mathbf{U}$ have the same column (range) space, we have $\mathbf{1}^\top \mathbf{U} = \mathbf{0}$. Moreover, as we construct $\bar{\mathbf{U}}$ such that $\mathbf{1}^\top \bar{\mathbf{U}} = \mathbf{0}$, we have $\mathbf{1}^\top [\mathbf{U} \ \bar{\mathbf{U}}] = \mathbf{0}$, which implies $\mathbf{1}^\top \mathbf{Y}^\star = \mathbf{0}$. Together with the fact that $(\mathbf{Y}^\star)^\top \mathbf{Y}^\star = n[\mathbf{V} \ \bar{\mathbf{V}}][\mathbf{U} \ \bar{\mathbf{U}}]^\top [\mathbf{U} \ \bar{\mathbf{U}}][\mathbf{V} \ \bar{\mathbf{V}}]^\top = n\mathbf{I}_r$, the feasibility of $\mathbf{Y}^\star$ to Eq. (6) does hold.

Now we consider an arbitrary $\mathbf{Y} \in \Omega$. Due to $\mathbf{1}^\top \mathbf{Y} = \mathbf{0}$, we have $\mathbf{JY} = \mathbf{Y} - \frac{1}{n}\mathbf{1}\mathbf{1}^\top \mathbf{Y} = \mathbf{Y}$, and $\langle \mathbf{B}, \mathbf{Y} \rangle = \langle \mathbf{B}, \mathbf{JY} \rangle = \langle \mathbf{JB}, \mathbf{Y} \rangle$. Moreover, by using von Neumann's trace inequality [2] and the fact that $\mathbf{Y}^\top \mathbf{Y} = n\mathbf{I}_r$, we have $\langle \mathbf{JB}, \mathbf{Y} \rangle \leq \sqrt{n} \sum_{k=1}^{r'} \sigma_k$. On the other hand,

$$\langle \mathbf{B}, \mathbf{Y}^\star \rangle = \langle \mathbf{JB}, \mathbf{Y}^\star \rangle$$

$$= \left\langle [\mathbf{U} \ \bar{\mathbf{U}}] \begin{bmatrix} \mathbf{\Sigma} & \mathbf{0} \\ \mathbf{0} & \mathbf{0} \end{bmatrix} [\mathbf{V} \ \bar{\mathbf{V}}]^\top, \mathbf{Y}^\star \right\rangle$$

$$= \sqrt{n} \left\langle \begin{bmatrix} \mathbf{\Sigma} & \mathbf{0} \\ \mathbf{0} & \mathbf{0} \end{bmatrix}, \mathbf{I}_r \right\rangle$$

$$= \sqrt{n} \sum_{k=1}^{r'} \sigma_k.$$

Therefore, we can quickly derive

$$\operatorname{tr}(\mathbf{B}^\top \mathbf{Y}) = \langle \mathbf{B}, \mathbf{Y} \rangle = \langle \mathbf{JB}, \mathbf{Y} \rangle$$

$$\leq \sqrt{n} \sum_{k=1}^{r'} \sigma_k$$

$$= \langle \mathbf{B}, \mathbf{Y}^\star \rangle$$

$$= \operatorname{tr}(\mathbf{B}^\top \mathbf{Y}^\star)$$

for any $\mathbf{Y} \in \Omega$, which completes the proof. $\quad\square$

**Theorem 1.** *If $\{(\mathbf{B}_k, \mathbf{Y}_k)\}$ is the sequence generated by Algorithm 2, then $\mathcal{Q}(\mathbf{B}_{k+1}, \mathbf{Y}_{k+1}) \geq \mathcal{Q}(\mathbf{B}_k, \mathbf{Y}_k)$ holds for any integer $k \geq 0$, and $\{\mathcal{Q}(\mathbf{B}_k, \mathbf{Y}_k)\}$ converges starting with any feasible initial point $(\mathbf{B}_0, \mathbf{Y}_0)$.*

*Proof.* Let us use $\|\cdot\|_2$ to denote matrix 2-norm. $\forall\, \mathbf{B} \in \{\pm 1\}^{n \times r}$ and $\forall\, \mathbf{Y} \in \Omega$, we can derive

$$
\begin{aligned}
\mathcal{Q}(\mathbf{B}, \mathbf{Y}) &= \langle \mathbf{B}, \mathbf{AB} \rangle + \rho \langle \mathbf{B}, \mathbf{Y} \rangle \\
&\leq \|\mathbf{B}\|_{\mathrm{F}} \|\mathbf{AB}\|_{\mathrm{F}} + \rho \|\mathbf{B}\|_{\mathrm{F}} \|\mathbf{Y}\|_{\mathrm{F}} \\
&\leq \|\mathbf{B}\|_{\mathrm{F}} \|\mathbf{A}\|_2 \|\mathbf{B}\|_{\mathrm{F}} + \rho \|\mathbf{B}\|_{\mathrm{F}} \|\mathbf{Y}\|_{\mathrm{F}} \\
&= \sqrt{nr} \cdot \sqrt{nr} + \rho \sqrt{nr} \cdot \sqrt{nr} = (1 + \rho)nr,
\end{aligned}
$$

where the second line is due to Cauchy-Schwarz inequality, the third line follows from the inequality $\|\mathbf{CD}\|_{\mathrm{F}} \leq \|\mathbf{C}\|_2 \|\mathbf{D}\|_{\mathrm{F}}$ for any compatible matrices $\mathbf{C}$ and $\mathbf{D}$, and the last line holds as $\|\mathbf{B}\|_{\mathrm{F}} = \|\mathbf{Y}\|_{\mathrm{F}} = \sqrt{nr}$ and $\|\mathbf{A}\|_2 = 1$. Since $(\mathbf{B}_k, \mathbf{Y}_k)$ is feasible at each $k$, the sequence $\{\mathcal{Q}(\mathbf{B}_k, \mathbf{Y}_k)\}$ is bounded from above.

Applying Lemma 1 and Lemma 2, we quickly have

$$
\mathcal{Q}(\mathbf{B}_{k+1}, \mathbf{Y}_{k+1}) \geq \mathcal{Q}(\mathbf{B}_{k+1}, \mathbf{Y}_k) \geq \mathcal{Q}(\mathbf{B}_k, \mathbf{Y}_k).
$$

Together with the boundedness of $\{\mathcal{Q}(\mathbf{B}_k, \mathbf{Y}_k)\}$, the monotonically non-decreasing sequence $\{\mathcal{Q}(\mathbf{B}_k, \mathbf{Y}_k)\}$ must converge. $\qquad\square$

**Proposition 1.** *For any orthogonal matrix $\mathbf{R} \in \mathbb{R}^{r \times r}$ and any binary matrix $\mathbf{B} \in \{\pm 1\}^{n \times r}$, we have $\mathrm{tr}(\mathbf{B}^\top \mathbf{AB}) \geq \dfrac{1}{r}\mathrm{tr}^2(\mathbf{R}^\top \mathbf{H}^\top \mathbf{AB})$.*

*Proof.* Since $\mathbf{A}$ is positive semidefinite, it suffices to write $\mathbf{A} = \mathbf{E}^\top \mathbf{E}$ with some proper $\mathbf{E} \in \mathbb{R}^{m' \times n}$ $(m' \leq m)$. Moreover, because the operator norm $\|\mathbf{A}\|_2 = 1$, we have $\|\mathbf{E}\|_2 = 1$.

By taking into account the fact $\mathbf{H}^\top \mathbf{H} = \mathbf{I}_r$, we can derive that for any orthogonal matrix $\mathbf{R} \in \mathbb{R}^{r \times r}$ and any binary matrix $\mathbf{B} \in \{\pm 1\}^{n \times r}$, the following inequality holds:

$$
\begin{aligned}
\left| \mathrm{tr}(\mathbf{R}^\top \mathbf{H}^\top \mathbf{AB}) \right| &= \left| \mathrm{tr}(\mathbf{R}^\top \mathbf{H}^\top \mathbf{E}^\top \mathbf{EB}) \right| \\
&= \left| \langle \mathbf{EHR}, \mathbf{EB} \rangle \right| \\
&\leq \|\mathbf{EHR}\|_{\mathrm{F}} \|\mathbf{EB}\|_{\mathrm{F}} \\
&\leq \|\mathbf{E}\|_2 \|\mathbf{HR}\|_{\mathrm{F}} \|\mathbf{EB}\|_{\mathrm{F}} \\
&= \sqrt{\mathrm{tr}(\mathbf{R}^\top \mathbf{H}^\top \mathbf{HR})} \|\mathbf{EB}\|_{\mathrm{F}} \\
&= \sqrt{\mathrm{tr}(\mathbf{I}_r)} \|\mathbf{EB}\|_{\mathrm{F}} \\
&= \sqrt{r} \|\mathbf{EB}\|_{\mathrm{F}}.
\end{aligned}
$$

The above inequality gives $\|\mathbf{EB}\|_{\mathrm{F}} \geq \left| \mathrm{tr}(\mathbf{R}^\top \mathbf{H}^\top \mathbf{AB}) \right| / \sqrt{r}$, leading to

$$
\begin{aligned}
\mathrm{tr}(\mathbf{B}^\top \mathbf{AB}) &= \mathrm{tr}(\mathbf{B}^\top \mathbf{E}^\top \mathbf{EB}) = \|\mathbf{EB}\|_{\mathrm{F}}^2 \\
&\geq \left( \left| \mathrm{tr}(\mathbf{R}^\top \mathbf{H}^\top \mathbf{AB}) \right| / \sqrt{r} \right)^2 \\
&= \frac{1}{r} \mathrm{tr}^2(\mathbf{R}^\top \mathbf{H}^\top \mathbf{AB}),
\end{aligned}
$$

which completes the proof. $\qquad\square$

It is also easy to prove the convergence of the sequence $\left\{ \mathrm{tr}\big((\mathbf{R}^j)^\top \mathbf{H}^\top \mathbf{AB}_0^j\big) \right\}_j$ generated in optimizing problem (8) of the main paper, by following the spirit of Theorem 1.

## 2 More Discussions

### 2.1 Orthogonal Constraint

Like Spectral Hashing (SH) [7, 6] and Anchor Graph Hashing (AGH) [4], we impose the orthogonal constraints on the target hash bits so that the redundancy among these bits can be minimized, which will lead to uncorrelated hash bits if the orthogonal constraints are strictly satisfied. However, the orthogonality of hash bits is traditionally difficult to achieve, so our proposed Discrete Graph Hashing (DGH) model pursues *nearly* orthogonal (uncorrelated) hash bits by softening the orthogonal constraints. Here we clarify that the performance drop of linear projection based hashing methods such as [1][3] is not due to the orthogonal constraints imposed on hash bits, but on the "projection directions". When using orthogonal projections, the quality of constructed hash functions typically degrades rapidly since most variance of the training data is captured in the first few orthogonal projections. On the contrary, in this work we impose the orthogonal constraints on hash bits. Even though recall is expected to drop with long codes for all the considered hashing techniques including our proposed DGH, orthogonal/uncorrelated hash bits have been found to yield better search accuracy (*e.g.*, precision/recall) for longer code lengths since they minimize the bit redundancy. Previous hashing methods [4, 6, 7] using continuous relaxations deteriorate with longer code lengths because of large errors introduced by the discretizations of their continuously relaxed solutions. Our hashing technique (both versions DGH-I and DGH-R) generates nearly uncorrelated hash bits via direct discrete optimization, so its recall decreases much slower and also keeps much higher in comparison to the other methods (see Figure 3). The hash lookup success rate of our DGH is kept almost 100% with only a tiny drop (see Figure 1 in the main paper). Overall, we have shown that the orthogonal constraints on hash bits, which are nearly satisfied by enforcing direct discrete optimization, do not hurt but improve recall/success rate of hash lookup for relatively longer code lengths.

### 2.2 Spectral Relaxation

Spectral methods have been well studied in the literature, and the effect of spectral relaxations which were widely adopted in clustering, segmentation, and hashing problems could be bounded. However, as those existing bounds are typically concerned with the worst case, such theoretical results may be very conservative without clear practical implications. Moreover, since we used the solution, obtained via continuous relaxation + discrete rounding, of the spectral method AGH [4] as an initial point for running our discrete method DGH-I, due to the proven monotonicity (Theorem 1), DGH (Algorithm 2 in the main paper) leads to a solution which is certainly no worse than the initial point (*i.e.*, the spectral solution) in terms of the objective function value of Eq. (4) in the main paper. Note that the deviation (*e.g.*, $\ell_1$ distance) from the continuously relaxed solution to the discrete solution will grow as the code length increases, as disclosed by the normalized cuts work [5]. Quantitatively, we find that the hash lookup F-measure (Figure 2 in the main paper) and recall (Figure 3) achieved by the spectral methods (*e.g.*, SH, 1-AGH and 2-AGH) relying on continuous relaxations either drop drastically or become very poor when the code length surpasses 48. Therefore, in the main paper, we argued that continuous relaxations applied for solving hashing problems may lead to poor hash codes with longer code lengths.

### 2.3 Initialization

Our proposed initialization schemes are motivated by previous work. In our first initialization scheme, $\mathbf{Y}_0$ originates from the spectral embedding of the graph Laplacian, and its binarization $\mathbf{B}_0$ (threshold $\mathbf{Y}_0$ at zero) has been used as the final hash codes by the spectral method AGH [4]. Our second initialization scheme modifies the first one to seek the optimally rotated spectral embedding $\mathbf{Y}_0$, where the motivation is provided by Proposition 1. While the two initialization schemes are heuristic, we find that both of them consistently result in much better performance than random initialization.

### 2.4 Out-of-Sample Hashing

We proposed a novel out-of-sample extension for hashing in Section 2 of the main paper, which is essentially discrete and completely different from the continuous out-of-sample extensions suggested by the spectral methods including SH [7], MDSH [6], 1-AGH and 2-AGH [4]. Our proposed

**(a) Convergence curves of Q(B,Y) with r=24**    **(b) Convergence curves of Q(B,Y) with r=48**    **(c) Convergence curves of Q(B,Y) with r=96**

Figure 1: Convergence curves of $\mathcal{Q}(\mathbf{B}, \mathbf{Y})$ starting with two different initial points on **CIFAR-10**.

**(a) Convergence curves of Q(B,Y) with r=24**    **(b) Convergence curves of Q(B,Y) with r=48**    **(c) Convergence curves of Q(B,Y) with r=96**

Figure 2: Convergence curves of $\mathcal{Q}(\mathbf{B}, \mathbf{Y})$ starting with two different initial points on **SUN397**.

extension neither makes any assumption nor involves any approximation, but directly achieves the hash code for any given query in a discrete manner. Consequently, this discrete hashing extension makes the whole hashing scheme *asymmetric* in the sense that hash codes are yielded differently for samples in the dataset and queries outside the dataset. In contrast, the overall hashing schemes of the above spectral methods [4, 6, 7] are all symmetric. We gave a brief discussion about the proposed asymmetric hashing mechanism in Section 4 of the main paper.

## 3    More Experimental Results

In Figures 1 and 2, we plot the convergence curves of $\left\{ \mathcal{Q}(\mathbf{B}_k, \mathbf{Y}_k) \right\}$ starting with the suggested two initial points $(\mathbf{B}_0, \mathbf{Y}_0)$. In specific, DGH-I uses the first initial point, and DGH-R uses the second initial point which is obtained by learning a rotation matrix $\mathbf{R}$. Note that if $\mathbf{R}$ is equal to the identity matrix $\mathbf{I}_r$, DGH-R degenerates to be DGH-I. Figures 1 and 2 indicate that when learning 24, 48, and 96 hash bits, DGH-R consistently gives a higher objective value $\mathcal{Q}(\mathbf{B}_0, \mathbf{Y}_0)$ than DGH-I at the start; DGH-R also reaches a higher objective value $\mathcal{Q}(\mathbf{B}^*, \mathbf{Y}^*)$ than DGH-I at the convergence, though DGH-R tends to need more iterations for convergence. Typically, more than one iteration are needed for both DGH-I and DGH-R to converge, where the first iteration usually gives rise to the largest increase in the objective function value $\mathcal{Q}(\mathbf{B}, \mathbf{Y})$.

We show the recall results of hash lookup within Hamming radius 2 in Figure 3, which reveal that when the number of hash bits $r \geq 16$, DGH-I achieves the highest recall except on **YouTube Faces**, where DGH-R is the highest while DGH-I is the second highest. Among those competing hashing techniques: when $r \geq 48$, KLSH, IsoH, SH and BRE suffer from poor recall ($< 0.065$), and even give close to zero recall on the first three datasets **CIFAR-10**, **SUN397** and **YouTube Faces**; ITQ attains much lower recall than DGH-I and DGH-R, and even gives close to zero recall when $r \geq 32$ on the first two datasets; LSH and IMH usually give higher recall than the other methods, but are notably inferior to DGH-I when $r \geq 16$; although using the same constructed anchor graph on each dataset, DGH-I and DGH-R achieve much higher recall than 1-AGH and 2-AGH which give close to zero recall when $r \geq 48$ on **CIFAR-10** and **Tiny-1M**.

Table 1: Hamming ranking performance on **CIFAR-10** and **SUN397** datasets. $r$ denotes the number of hash bits used in the hashing methods. All training and test times are in seconds.

| Method | CIFAR-10 | | | | | SUN397 | | | | |
|---|---|---|---|---|---|---|---|---|---|---|
| | Mean Average Precision | | | TrainTime | TestTime | Mean Average Precision | | | TrainTime | TestTime |
| | $r=24$ | $r=48$ | $r=96$ | $r=96$ | $r=96$ | $r=24$ | $r=48$ | $r=96$ | $r=96$ | $r=96$ |
| $\ell_2$ Scan | 0.1752 | | | | – | 0.1550 | | | | – |
| LSH | 0.1220 | 0.1228 | 0.1256 | 0.5 | $1.1\times10^{-5}$ | 0.0178 | 0.0171 | 0.0228 | 0.9 | $1.1\times10^{-5}$ |
| KLSH | 0.1275 | 0.1361 | 0.1407 | 1.8 | $3.7\times10^{-5}$ | 0.0404 | 0.0641 | 0.0842 | 5.2 | $4.8\times10^{-5}$ |
| ITQ | 0.1714 | 0.1783 | 0.1828 | 14.1 | $1.2\times10^{-5}$ | 0.1050 | 0.1276 | 0.1463 | 18.7 | $1.1\times10^{-5}$ |
| IsoH | 0.1651 | 0.1721 | 0.1765 | 1.8 | $1.2\times10^{-5}$ | 0.0883 | 0.0955 | 0.1218 | 9.2 | $1.1\times10^{-5}$ |
| SH | 0.1317 | 0.1352 | 0.1296 | 9.2 | $1.0\times10^{-4}$ | 0.0727 | 0.0791 | 0.0909 | 17.0 | $9.8\times10^{-5}$ |
| MDSH | 0.1616 | 0.1637 | 0.1631 | 13.3 | $7.2\times10^{-5}$ | 0.0823 | 0.0984 | 0.1134 | 54.8 | $1.4\times10^{-4}$ |
| IMH | 0.1832 | 0.1878 | 0.1925 | 15.4 | $2.6\times10^{-5}$ | 0.0987 | 0.1049 | 0.1102 | 35.2 | $3.5\times10^{-5}$ |
| 1-AGH | 0.1805 | 0.1685 | 0.1522 | 12.9 | $3.4\times10^{-5}$ | 0.1411 | 0.1486 | 0.1493 | 32.4 | $4.3\times10^{-5}$ |
| 2-AGH | 0.1812 | 0.1842 | 0.1719 | 13.5 | $5.5\times10^{-5}$ | 0.1256 | 0.1437 | 0.1544 | 33.5 | $5.2\times10^{-5}$ |
| BRE | 0.1619 | 0.1644 | 0.1713 | 943.9 | $5.6\times10^{-5}$ | 0.0683 | 0.0898 | 0.1096 | 1344.9 | $4.5\times10^{-5}$ |
| **DGH-I** | 0.1808 | 0.1819 | 0.1832 | 46.0 | $3.3\times10^{-5}$ | 0.1219 | 0.1119 | 0.1069 | 87.5 | $3.5\times10^{-5}$ |
| **DGH-R** | **0.1910** | **0.1912** | **0.1950** | 73.1 | $3.3\times10^{-5}$ | **0.1438** | **0.1575** | **0.1624** | 127.7 | $3.5\times10^{-5}$ |

From the recall results, we can conclude that the discrete optimization procedure exploited by our DGH hashing technique better preserves the neighborhood structure inherent in massive data into a discrete code space, than the relaxed optimization procedures employed by SH, 1-AGH and 2-AGH. We argue that the spectral methods SH, 1-AGH and 2-AGH did not really minimize the Hamming distances between the neighbors remaining in the input space, since the discrete binary constraints which should be imposed on the hash codes were discarded.

Finally, we report the Hamming ranking results on **CIFAR-10** and **SUN397** in Table 1, which clearly show the superiority of DGH-R over the competing hashing techniques in mean average precision (MAP). Regarding the reported training time, most of the competing methods such as AGH are non-iterative, while BRE and our proposed DGH-I/DGH-R involve iterative optimization procedures so they are slower. Admittedly, our DGH method is slower in training than the other methods except BRE, and DGH-R is slower than DGH-I due to the longer initialization for optimizing the rotation **R**. However, considering that training happens in the offline mode and substantial quality improvements are found in the learned hash codes, we believe that a modest sacrifice in training time is acceptable. Note that in Table 1 of the main paper, even for one million samples of the Tiny-1M dataset, the training time of DGH-I/DGH-R is less than one hour. Search/query time of all referred hashing methods includes two components: coding time and table lookup time. Compared to coding time, the latter is constant (dependent on the number of hash bits $r$) and small enough to be ignored, since a single hash table with no reordering is used for all the methods. Hence, we only report the coding time as the test time per query for all the compared methods. In most information retrieval applications, test time is usually the main concern as search mostly happens in the online mode, and our DGH method has comparable test time with the others.

To achieve the best performance for DGH-I/DGH-R, we need to tune the involved (hyper)parameters via cross validation. However, in our experience they are normally easy to tune. Regarding the parameters needed by anchor graph construction, we just fix the number of anchors $m$ as 300 as in the AGH method [4], fix $s$ to 3, and set the other parameters according to [4], so as to make a fair comparison with AGH. Regarding the budget iteration numbers required by the alternating maximization procedures of our DGH method, we simply fix them to proper constants (*i.e.*, $T_R = 100$, $T_B = 300$, $T_G = 20$), within which the initialization procedure for optimizing the rotation **R** and the DGH procedure (Algorithm 2 in the main paper) empirically converge. For the penalty parameter $\rho$ that controls the balancing and uncorrelatedness of the learned hash codes, we tune it in the range of $[0.1, 5]$ on each dataset. Our choices for these parameters are found to work reasonably well over all datasets.