[Reviews · NeurIPS 2014]

Submitted by Assigned_Reviewer_4

This work proposes a novel objective function and associated algorithm for the learning based hashing problem, and provides empirical evidence of its effectiveness. The paper is well written, the new objective is intuitive, and the experiments are convincing, though given that the experiments are the main source of evidence supporting this algorithm, they should nonetheless be improved.
Summary: The paper is long / dense (though well written on the whole). The authors should consider shortening it to make it more readable. The intro in particular can be condensed.

Till -> Until

While continuous relaxations have their drawbacks, they are theoretically sound, as the relaxed problems are typically convex or spectral, and the effect of the relaxation can be bounded. In contrast, the mixed integer program proposed in this work is NP-hard, and DGH converges to a local optimum and is thus less theoretically sound (though it admittedly seems to work well in their experiments). The authors need to do a better job of presenting these tradeoffs in a more balanced manner instead of simply referring to the relaxed problems as ‘error prone’.

Definition of neighbors. It would be interesting to show results both for neighbors defined based on class label and also based on l2 distances. Moreover, it’s not clear that top 2% makes sense for defining a neighbor, as 20K neighbors seems like a lot of neighbors. More generally, it would be good to know how sensitive the empirical results are to these definitions of neighbors.

Are the reported results generated via the test dataset? If so, did the experiments involve a single test/train split? Given how close some of the algorithms are for some of the results (in particular AGH in Table 1), it would be good to provide error bars on the results.

The authors should add more comments on the differences in training time. For instance, AGH 4.5 times faster than DGH at training time. Does this matter?

Submitted by Assigned_Reviewer_14

Summary:

The paper studies hashing-based search in large databases. Unsupervised methods have been recently used to produce more compact hashes than their randomized equivalents. The authors demonstrate that existing methods suffer from bad performance as the length of the codes increases and suggest a new graph-based method. To achieve better codes, they keep the binary constraints and consider a slightly relaxed formulation (still NP-hard) that they solve using alternating maximization. Local convergence is guaranteed and extensive experiments show that the suggested method achieves very good performance on a number of datasets.

Pros:

Very well written. Good, informative introduction with many well-organized references.

Many of the steps in the derivation come with good intuition and some novelty and the final algorithm is relatively simple and effective.

Assuming potential performance issues can be addressed (see my comment below) this could potentially be very impactful.

Cons:

1. One concern about the suggested method is computational complexity and scalability. In your experiments, we see that your algorithm gives solutions of good quality but is also relatively slow (only BRE was slower in Table 1). Is this an issue in practice? Does it get even worse with larger problem size? Can it be addressed somehow?
Summary: Very well written and complete paper on a good idea. It should be accepted.

Submitted by Assigned_Reviewer_37

Summary: This paper proposes an approach to learn r-bit binary hash codes for a given dataset, that are smooth with respect to an affinity graph. Unlike spectral hashing, the method does not relax the binary-value constraints on the hash codes upfront, and uses an approximate affinity graph based on a small subset of anchor points. After some transformations, the resulting optimization problem is solved using alternating updates, for which monotonic descent and convergence is established.

Pros:
- Detailed empirical comparisons on multiple datasets against a large set of alternative hashing techniques show that the proposed method consistently performs best.
- Despite being a nasty-looking discrete optimization problem, the authors are able to derive relatively simple alternating update rules for which some notion of convergence can apparently be established.

Cons:
- The algorithm has a large number of hyper-parameters (excluding bit length): s,m in Anchor graph construction, rho, and number of iterations in the alternating updates. It is not really made clear how the performance varies with respect to choices for these hyper parameters for a given new problem.
- The notion of convergence in Theorem 1 is not made clear. A proof should be sketched.
- Ultimately, the method is based on non-convex optimization and the quality of the solution cannot be guaranteed.
Summary: Good, practical paper; solid empirical results.
Author Feedback
Author rebuttal: We thank the reviewers for their constructive comments. Below we address their main concerns.

Reviewer_14
Q1. About computational complexity and scalability.
A1. Referring to the first paragraph of Section 4, the training time complexity of our method (including two versions DGH-I and DGH-R) is linear in the size of the training set, and the test time complexity is constant independent of the dataset size. Hence, our method is easy to scale up to massive datasets. Admittedly, our method is slower at training than other methods (except BRE). But considering that training happens in the offline mode, and also the substantial quality improvement in the learned hash codes, we believe that a modest sacrifice in the training time is acceptable (note that in Table 1, even for 1 million samples of Tiny-1M, the training time is well under an hour). In most information retrieval applications, test time is usually the main concern as search happens in the online mode, and our method has similar test time to others.

Reviewer_37
Q1. “The algorithm has a large number of hyper-parameters…how the performance varies with respect to choices for these hyper-parameters”.
A1. To achieve the best performance, we need to tune these (hyper-)parameters via cross validation. However, in our experience they are normally easy to tune. Regarding the parameters in anchor graph construction, we just fix the number of anchors m as 300 as in [23], fix s to 3, and set the other parameters according to [23], so as to make a fair comparison with [23]. Regarding the budget iteration numbers used in alternating optimization, we also simply fix them to proper constants, within which the updating procedures converge. For the penalty parameter $\rho$, we tune it in the range of [0.1,5] on each dataset. Our choices for these parameters were found to work reasonably well over all datasets. In the supplementary material, we will add the performance changes of our method with respect to varying parameters, e.g., the changes of search precision/recall w.r.t. varying $\rho$ that controls the balancing and uncorrelatedness of the learned hash codes.

Q2. “The notion of convergence in Theorem 1 is not made clear. A proof should be sketched”.
A2. The notion of convergence in Theorem 1 is with regard to the objective in Eq. (4). Due to the nature of our proposed alternating maximization procedure, Alg. 2 always generates a monotonically non-decreasing and upper bounded sequence of objective values, and thus converges. We have given a complete proof to Theorem 1 in the sup-material, and will sketch the proof in the final paper.

Reviewer_4
Q1. “Presenting these tradeoffs in a more balanced manner instead of simply referring to the relaxed problems as ‘error prone’ ”.
A1. We agree that convex/spectral methods are better studied and the effect of spectral relaxations can be bounded. However, as those bounds are typically concerned with the worst case, the results can be very conservative without clear practical implications. Moreover, since we used the solution of the spectral method [23] as the initial point for discrete optimization (see lines 270-275), due to the monotonicity (Theorem 1), Alg. 2 leads to a solution which is for sure no worse than the initial point (i.e., the spectral solution) in terms of the objective value of Eq. (4). We will highlight the above points in the revised version.

Q2. About definitions of neighbors.
A2. Since the first three datasets CIFAR-10, SUN397 and YouTube Faces all have ground-truth class labels, we adopted the conventional semantic neighbor definition, i.e., two samples sharing the same class label are considered as neighbors, for fair comparison with other techniques. The last dataset Tiny-1M is not annotated, so we did the experiments based on the L2 neighbor definition. Top 2% is just an empirical setting for defining L2-distance neighbors, like previous works [22,35]. We can certainly modify the neighbor definition (e.g., top 0.1%, 0.5%, and 1%), and present the experimental results accordingly.

Q3. About the reported results.
A3. All of the reported results are obtained over test samples, i.e., queries. We stated the training/test split for each tried dataset in lines 357-363, which followed previous works. We will add error bars for each reported precision in Table 1. Regarding the reported training time, most of the competing methods such as AGH are noniterative, while BRE, DGH-I and DGH-R involve iterative optimization procedures so they are slower (also see Q1/A1 to Rev14). We will add more comments on training time in the revised version.